# Magnetic Resonance Imaging-Based Predictive Models for Clinically Significant Prostate Cancer: A Systematic Review

**DOI:** 10.3390/cancers14194747

**Published:** 2022-09-29

**Authors:** Marina Triquell, Miriam Campistol, Ana Celma, Lucas Regis, Mercè Cuadras, Jacques Planas, Enrique Trilla, Juan Morote

**Affiliations:** 1Department of Urology, Vall d’Hebron University Hospital, 08035 Barcelona, Spain; mcampistol@vhebron.net (M.C.); acelma@vhebron.net (A.C.); lregis@vhebron.net (L.R.); mcuadras@vhebron.net (M.C.); jplanas@vhebron.net (J.P.); etrilla@vhebron.net (E.T.); juan.morote@vallhebron.cat (J.M.); 2Department of Surgery, Univesirtat Autònoma de Barcelona, 08193 Barcelona, Spain

**Keywords:** prostate cancer, magnetic resonance imaging, predictive model, risk calculator

## Abstract

**Simple Summary:**

Magnetic resonance imaging (MRI) has allowed the early detection of PCa to evolve towards clinically significant PCa (csPCa), decreasing unnecessary prostate biopsies and overdetection of insignificant tumours. MRI identifies suspicious lesions of csPCa, predicting the semi-quantitative risk through the prostate imaging report and data system (PI-RADS), and enables guided biopsies, increasing the sensitivity of csPCa. Predictive models that individualise the risk of csPCa have also evolved adding PI-RADS score (MRI-PMs), improving the selection of candidates for prostate biopsy beyond the PI-RADS category. During the last five years, many MRI-PMs have been developed. Our objective is to analyse the current developed MRI-PMs and define their clinical usefulness through a systematic review. We have found high heterogeneity between MRI technique, PI-RADS versions, biopsy schemes and approaches, and csPCa definitions. MRI-PMs outperform the selection of candidates for prostate biopsy beyond MRI alone and PMs based on clinical predictors. However, few developed MRI-PMs are externally validated or have available risk calculators (RCs), which constitute the appropriate requirements used in routine clinical practice.

**Abstract:**

MRI can identify suspicious lesions, providing the semi-quantitative risk of csPCa through the Prostate Imaging-Report and Data System (PI-RADS). Predictive models of clinical variables that individualise the risk of csPCa have been developed by adding PI-RADS score (MRI-PMs). Our objective is to analyse the current developed MRI-PMs and define their clinical usefulness. A systematic review was performed after a literature search performed by two independent investigators in PubMed, Cochrane, and Web of Science databases, with the Medical Subjects Headings (MESH): predictive model, nomogram, risk model, magnetic resonance imaging, PI-RADS, prostate cancer, and prostate biopsy. This review was made following the Preferred Reporting Items for Systematic Reviews and Meta-analyses (PRISMA) criteria and studied eligibility based on the Participants, Intervention, Comparator, and Outcomes (PICO) strategy. Among 723 initial identified registers, 18 studies were finally selected. Warp analysis of selected studies was performed with the Quality Assessment of Diagnostic Accuracy Studies (QUADAS-2) tool. Clinical predictors in addition to the PI-RADS score in developed MRI-PMs were age, PCa family history, digital rectal examination, biopsy status (initial vs. repeat), ethnicity, serum PSA, prostate volume measured by MRI, or calculated PSA density. All MRI-PMs improved the prediction of csPCa made by clinical predictors or imaging alone and achieved most areas under the curve between 0.78 and 0.92. Among 18 developed MRI-PMs, 7 had any external validation, and two RCs were available. The updated PI-RADS version 2 was exclusively used in 11 MRI-PMs. The performance of MRI-PMs according to PI-RADS was only analysed in a single study. We conclude that MRI-PMs improve the selection of candidates for prostate biopsy beyond the PI-RADS category. However, few developed MRI-PMs meet the appropriate requirements in routine clinical practice.

## 1. Introduction

Prostate cancer (PCa) suspicion, based on elevated serum prostate-specific antigen (PSA) and/or abnormal digital rectal examination (DRE), followed by systematic prostate biopsy, has been the classic strategy for early detection of PCa [1]. However, this strategy has led to a high rate of unnecessary prostate biopsies and overdetection of insignificant PCa (iPCa) [2]. Currently, early detection of PCa has evolved towards the detection of clinically significant PCa (csPCa), which is usually defined in prostate biopsies as the International Society of Uro-Pathology (ISUP) grade group (GG) 2 or higher [3]. However, other existing definitions of csPCa based on tumour burden and PSA density (PSAD) in addition to ISUP GG may correlate better with the histopathological findings of surgical specimens [4].

Most relevant progress in the early detection of csPCa has come from pre-biopsy multiparametric magnetic resonance imaging (mpMRI), which can identify suspicious lesions, providing the semi-quantitative risk of csPCa through the Prostate Imaging-Report and Data System (PI-RADS) [5]. Additionally, guided biopsies of suspicious lesions identified in mpMRI increase the sensitivity of csPCa reached with systematic biopsies [6,7]. The negative predictive value of mpMRI for csPCa reaches up to 95% [8]; therefore, a prostate biopsy is usually avoided in men with PI-RADS <3 [9] However, the positive predictive value of mpMRI ranges from 15 to 20% in men with PI-RADS 3, between 50–60% in those with PI-RADS 4, and between 85–95% in men with PI-RADS 5 [10]. Consequently, there are uncertain scenarios in which modern markers, PSAD, and predictive models based on MRI findings and clinical predictors (MRI-PMs) are re-commended for improving the selection of candidates for prostate biopsy [11,12,13].

Predictive models (PMs) are the only tool providing the individual risk of csPCa. These PMs have usually been presented through nomograms [14]; however, the cumbersome and time-consuming use of nomograms makes designing user-friendly web and smartphone available risk calculators (RCs) essential to facilitating the daily use of PMs [15,16,17]. In addition, PMs need external validation in the populations where they will be used to know if predictions are accurate. From the spread of pre-biopsy MRI, many MRI-PMs have been developed; however, the heterogenicity of development cohorts, diffe-rences between MRI scanners and PI-RADS versions used, variability of prostate biopsy schemes and approaches, and different csPCa definitions, in addition to external validations and availability of RCs, justify a systematic review. This systematic review analyses the current developed MRI-PMs and defines their clinical usefulness.

## 2. Evidence Acquisition

### 2.1. Search Strategy

Two independent reviewers (M.T. and J.M.) searched the literature to retrieve all relevant studies published before 31 March 2022. This search was conducted in PubMed, Cochrane, and Web of Science databases, using the Medical Subject Headings (MeSH): nomogram, risk model, magnetic resonance imaging, PI-RADS, prostate cancer, and prostate biopsy. The Preferred Reporting Items for Systematic Reviews and Meta-analyses (PRISMA) criteria [18] were followed. Disagreements were resolved through discussion between reviewers. This systematic review was registered in PROSPERO (International prospective register of systematic reviews), with the ID number CRD42022345848.

### 2.2. Eligible Criteria

The study eligibility was based on the Participants, Intervention, Comparator, and Outcomes (PICO) strategy [19]. The inclusion criteria were (1) the population at risk of PCa due to abnormal DRE or elevated serum PSA; (2) MRI identification of suspicious lesions and the performance of guided and systematic biopsies; (3) a comparison between PMs based on clinical parameters alone and MRI; (4) the outcome of csPCa. The search criteria were limited to articles written in English. Review articles, meta-analyses, conference abstracts, and letters were excluded.

### 2.3. Study Selection

The flow diagram of study selection is illustrated in Figure 1. Among 723 registers initially identified, 302 were duplicated. The retrieved study abstracts were screened and selected by exclusion and inclusion criteria, with 267 excluded. Of the 154 remaining stu-dies, 136 were excluded after full-text screening because the outcomes were not the pre-diction of csPCa before a prostate biopsy, or a PM was not developed. Finally, 18 studies fulfilled the eligibility criteria.

### 2.4. Quality Assessment

Quality Assessment of Diagnostic Accuracy Studies 2 (QUADAS-2) was used as an evidence-based quality assessment tool [20]. QUADAS-2 comprised four domains: patient selection, index test, reference standard, and flow and timing. The risk of bias in each study was evaluated by two independent reviewers as low, high, or unclear. The QUADAS-2 results are summarised in Figure 2, suggesting an overall low risk of bias.

## 3. Synthesis of the Evid

The characteristics of development cohorts of each MRI-PM regarding size, prostate biopsy status (initial vs. repeat), MRI characteristics and reporting methods, prostate biopsy scheme and approach, types of guided biopsies, and the definition of csPCa are summarised in Table 1. The logistic regression analysis among the selected candidate clinical predictors of csPCa, in addition to the MRI report, was the only method used to generate all MRI-PMs. The independent clinical predictors of csPCa were included in each MRI- PM and are presented in Table 2. The csPCa discrimination ability of MRI-PMs was analysed using areas under the receiver operating characteristic curves (AUROCs). The net benefit over biopsy of all the men or other predictors as baseline models of clinical predictors or MRI alone were examined through decision curve analysis (DCAs). The clinical utility was explored with clinical utility curves (CUCs) as a representation of avoided biopsies and missed csPCa in a continuous evolution of thresholds for csPCa risk.

In addition, the specificity corresponded to 95% sensitivity of each MRI-PM and avoided prostate biopsies are summarised in Table 3. Below we review the most significant characteristics and findings of developed MRI-PMs regarding the year of publication.

In 2016, Fang et al. [21] developed the first MRI-PM. They recruited 984 biopsy-naïve men from a single centre in Pekin, China, in whom guided and 12-core systematic biopsies were performed by transrectal approach after multiparametric MRI (mpMRI) between 2011 and 2013. The CsPCa (GS > 3+4) detection was 24.4%. Age, DRE, serum PSA, and prostate volume (PV) assessed from the transrectal ultrasound (TRUS) just before prostate biopsy, were the clinical predictors, in addition to the retrospective review of PI-RADS v1 as grade 0 (PI-RADS 1 and 2), grade 1 (PI-RADS 3), and grade 3 (PI-RADS 4 and 5). The AUROC of MRI-PM compared to that of the baseline model of clinical predictors were 0.87 and 0.85, respectively (*p* = 0.001). DCAs suggested the net benefit of MRI-PM over MRI and biopsy of all men from the risk threshold of 5%. At 95% sensitivity, the specificity was 38%, and the percentage of avoided biopsies was 19.8%. This model was not externally validated, and no RC was designed. Also in 2016, Kim et al. [22] incorporated MRI into the Prostate Cancer Prevention Trial RC (PCPT-RC). The MRI-PM was developed in 339 suspected PCa men (185 biopsy-naïve and 154 men subjected to repeat biopsy) from St. Louis, US, in whom 34% csPCa (GS ≥ 3+4) was detected after transrectal cognitive or -guided biopsies and 12-core systematic biopsy carried out between 2012 and 2015. MRI-PM improved the discrimination ability of the PCPT-RC of csPCa but not significantly; the AUROCs were 0.78 and 0.74, respectively (*p* = 0.06). At 95% sensitivity, the specificity was 20% and the rate of avoided biopsies was 15.1%. This MRI-PM was not externally validated and the current available PCPT-RC v.2.0 does not incorporate the MRI findings.

In 2017, six MRI-PMs were developed. Bjurlin et al. [23] developed two MRI-PM for csPCa (GS ≥ 3+4) regarding the biopsy status of 464 suspected PCa men, 288 biopsy-naïve and 171 with prior negative prostate biopsy. Prebiopsy mpMRI and 1 to 4-core software-guided biopsies and/or 12-core systematic biopsy carried out by transrectal approach were performed between 2012 and 2014, in New York, US. The rates of csPCa detection were 33.6% and 18.1%, respectively. The MRI-PM included age, PSAD, and PI-RADS v1 score for the mpMRI report as predictors. The AUROCs for csPCa detection were 0.91 in biopsy- naïve men and 0.86 in those with previous negative biopsies; 42% and 34% of prostate biopsies were avoided in these groups, missing 5% of csPCa. These models were not externally validated, and RC was not designed. Lee et al. [24] developed an MRI-PM including age, biopsy status, PSAD, and biparametric MRI (bpMRI) as predictors of csPCa (GS > 7 or maximal core cancer length > 6 mm) in a cohort of 615 suspected. PCa men (21.3% with prior negative prostate biopsy) subjected to transperineal template biopsy (24–40 cores) and 2- to 5-core cognitive-guided biopsies in Essex, UK, between 2012 and 2015. The rate of detected csPCa was 38.5%. The AUROC of MRI improved from 0.87 to 0.92. With the 30% csPCa risk threshold, 34.6% of prostate biopsies were avoided, and 2.5% of csPCa were missed. No external validations were made, and no RC was designed. Niu et al. [25] developed an MRI-PM for csPCa (GS > 3+4) among 151 biopsy-naïve men with serum PSA between 4 and 10 ng/mL. After mpMRI, a transrectal 1 or more-core cognitive-guided biopsy of suspicious lesions and/or a 12-core systematic biopsy were performed between 2014 and 2015 in Chengdu, China. The detection rate of csPCa was 21%, and the predictors of the model were age, PSAD, and PI-RADS v2. The AUROC of MRI increased from 0.76 to 0.85 when the model included the age and PSAD. The MRI-PM with 87.3% sensitivity obtained from the 36% csPCa threshold avoided 64.9% of prostate biopsies. The model was internally validated in a subsequent series of 74 men in the same centre; however, it was not externally validated, and no RC was designed. Radtke et al. [26] developed two MRI-PMs of csPCa (GS > 3+4) after sharing the same clinical predictors used by the Rotterdam RC-3 and RC-4 among 1159 consecutive suspected PCa men subjected to mpMRI reported by PI-RADS v1, and transperineal 24-core template biopsies and 2 to 4 additional cognitive-guided biopsies to suspicious lesions, between 2012 and 2015, in Heidelberg, Germany. The rates of detected csPCa were not provided; however, the model developed from 660 biopsy-naïve men reached an AUC of 0.83 compared to the 0.81 obtained from 335 men with previous negative prostate biopsy. Both models presented a net benefit over mpMRI from 16 and 12% risk of csPCa, respectively. Additionally, their specificities were 35% and 25.5%, respectively, at 95% sensitivity. The rates of avoided biopsies and missed csPCa were not provided. These models have not been externally validated and no RC has been designed. Truong et al. [27] developed an MRI-PM in 285 men with suspected PCa and at least one previous negative biopsy from two institutions in Rochester, NY, and Birmingham, Alabama, US. Pre-biopsy mpMRI and 12- to 24-core transrectal biopsies and 2-core software-guided biopsies to suspicious lesions were performed, and 38.9% of csPCa (GS > 3+4) was detected. The predictors shared in the model were age, serum PSA, PV, and PI-RADS v2. The AUROC to discriminate csPCa was 0.83; with a 40% threshold, 36.5% of prostate biopsies were avoided while missing 6.3% of csPCa. The model exhibited a net benefit over MRI from the 3% risk threshold. External validation of the Truong’s MRI-PM was carried out by Bjurlin et al. [36] in a cohort of 104 men with previous negative prostate biopsy in whom csPCa was detected in 49%. The AUROCs were 0.77 and 0.80, respectively. In the validation cohort, 55% of prostate biopsies were avoided missing 7% of csPCa. RC has not been designed. Finally, van Leeuwen et al. [28] also reported their MRI-PM in 2017. The model was developed in 393 suspected PCa men in whom pre-biopsy mpMRI and transperineal 30-core template prostate biopsy with 2 additional-core software or cognitive-guided biopsies of each suspicious lesion were performed between 2012 and 2014 in Sidney, Australia. The model included age, DRE, serum PSA, PV, and PI-RADS v1 as predictors, and the rate of detected csPCa (GS > 7 and >5% of G 4, and maximal core cancer length > 20% or 7 mm) was 37.9%. The baseline model with clinical predictors presented an AUROC of 0.80, increasing to 0.88 when the MRI was shared (*p* < 0.001). A threshold of 12.5% of csPCa avoided 34.3% of prostate biopsies while missing 6.1% of csPCa. The MRI-PM exhibited a net benefit over MRI and biopsy of all men from the csPCa risk of 3%. This model was externally validated in a cohort of 198 men, from another hospital in Sidney, with an AUROC of 0.86. No RC was designed.

In 2018, Huang et al. [30] developed an MRI-PM including age, serum PSA, DRE, PV measured from TRUS, and mpMRI reported by PI-RADS v2 in 231 suspected PCa men after an initial negative prostate biopsy performed between 2007 and 2017, in whom 2-core guided biopsies to suspicious lesions and/or 12-core systematic biopsies were carried out by transrectal approach in Beijing, China. The detection rate of csPCa (GS > 3+4) was 25.5%. The AUROC of MRI-PM increased to 0.92 from the 0.88 of the baseline model of clinical predictors. DCA showed a net benefit of MRI-PM over the baseline model from the csPCa risk threshold of 10%. At 95% sensitivity of the MRI-PM, the specificity was 63%, and 48% of prostate biopsies were avoided. No external validation of this MRI-PM was made, and RC was not designed. Also in 2018, Mehralivand et al. [31] developed an MRI-PM in 400 suspected PCa, 55.2% of them with previous negative prostate biopsy. After mpMRI transrectal 2-core software-guided biopsy and/or 12-core systematic biopsy were performed between 2015 and 2016 in Bethesda, US. The rate of detected csPCa (GS > 3+4) was 48.3%. The MRI-PM, including DRE, biopsy status, ethnicity, PSA, and PI-RADS v2, exhibited an AUROC of 0.84 compared to 0.72 of the baseline model without MRI (*p* < 0.001). At 96% sensitivity of the MRI-PM, the specificity was 54%, and 30% of prostate biopsies were avoided. DCA showed the net benefit of MRI-PM over the baseline model and MRI alone from the 10% risk threshold of csPCa. This MRI-PM was externally validated in 251 men from two institutions in Chicago, IL and Birmingham, AL, where the incidence of csPCa was 41.6 and 36%, respectively. No RC was designed.

In 2019, Boesen et al. [32] developed a PM of csPCa (GG > 2) including bpMRI, reported by PI-RADS v2 and stratified as negative (score 1–2), equivocal (score 3), and po- sitive (score 4–5), in addition to age, DRE, and PSAD, in 876 biopsy-naïve men from Herlev, Denmark. Suspected PCa men over 75 years or PSA serum levels behind 50 ng/mL were excluded. These men were subjected to transrectal 2-core cognitive-guided biopsies for each suspicious lesion and/or 10-core systematic biopsies between 2015 and 2017. The AUROC of 0.85, obtained from the baseline model with clinical predictors, increased to 0.89 when the MRI was shared (*p* < 0.001). The net benefit of MRI-PM over MRI alone, baseline model, and biopsy of all men were observed from a 5% risk of csPCa. At 96% sensitivity, the MRI-PM presented a specificity of 60%, and 38% of prostate biopsies were avoided. No external validation was performed, and no RC was designed. Alberts et al. [29] adjusted the Rotterdam RC-3 and RC-4 to a multi-centre population of 961 suspected PCa men, 504 biopsy-naïve and 457 men with prior negative biopsy, in whom csPCa (GS > 3+4) was detected in 42% and 29% respectively. These two models incorporated prebiopsy MRI, mostly multiparametric and reported by PI-RADS v1. Transrectal cognitive, software, or in bore-guided biopsies and/or 12-core systematic biopsies were performed, between 2012 and 2017, in Düsseldorf and four Dutch cities. The Rotterdam RC-3 improved its AUROC from 0.76 to 0.84 when the MRI was shared, while the RC-4 improved from 0.74 to 0.85. DCAs showed a net benefit of MRI-PM over RC-3 and RC-4 from the respective 10% and 4% csPCa risk thresholds. At 92% sensitivity, 24% of prostate biopsies were avoided in biopsy-naïve men, and at 96% sensitivity, 41% of prostate biopsies were avoided in men with prior negative prostate biopsy. Both MRI-PMs have been incorporated into the current Rotterdam MRI-RC, which has been externally validated in populations from China, Italy, and the UK [37,38,39]. Finally, Borque-Fernando et al. [15] developed an MRI-PM to predict csPCa (GG > 2) in 346 men with suspected PCa, 52.8% with prior negative prostate biopsy in whom transrectal 2-core cognitive-guided biopsies to suspicious lesions and/or 12-core systematic biopsy were performed between 2015 and 2016 in Barcelona, Spain. The rate of detected csPCa was 32.6%. Age, DRE, PSAD, biopsy status, and PI-RADS v2 were the individual predictors of csPCa included in the model. The AUROC of MRI-PM was 0.88, and CUCs showed at 10% csPCa risk threshold that 30% of prostate biopsies were avoided, while 95% of csPCa were de-tected. This model was not externally validated, and no RC is available.

In 2020, Chen et al. [33] developed an MRI-PM to discriminate between low-grade PCa (GS < 6) and high-grade (GS > 3+4) among 257 newly diagnosed PCa between 2017 and 2019 in Wuhan, China. PCa was diagnosed after prebiopsy mpMRI and no referred type and approach of prostate biopsy. The rate of high-grade PCa was 59.2%. This PM shared serum PSA, PV, and PI-RADS v.2. The AUROC was 0.84, and the sensitivity and specificity of the greatest discriminative threshold were 79.4% and 77.6%, respectively. A median serum PSA of 51 ng/mL was noted in this development cohort. The corresponding sensitivity at 95% specificity of the model was 40%, and 19% of overdetection of low-grade PCa would be avoided. The net benefit of this MRI-PM developed was not analysed, and external validation of the designed nomogram was carried out on 57 patients in Xiangyang, China. The sensitivity to predict high-grade PCa in this validation cohort was 80%, the specificity was 77%, and the correlation between the prediction threshold and high-grade PCa was 57%. Noh et al. [34], also in 2020, developed an MRI-PM to predict csPCa (GS > 3+4), sharing age, PSAD and bpMRI reported with PI-RADS v2 in 300 suspected PCa men, with 28.3% having a previous negative prostate biopsy, from Seoul, Korea. These men were subjected to transperineal up to 10-core software-guided prostate biopsies to PI-RADS > 3 lesions and/or from 14 to 20-core template-prostate biopsy between 2017 and 2019. The csPCa detection rate was 34%. The AUROC of the MRI-PM model was 0.86 compared to 0.79 of the clinical predictors model. The MRI-PM resulted in a net benefit over MRI and the clinical predictors model from a 10% csPCa threshold. The 95% sensitivity threshold provided 52% specificity and avoided 30% of prostate biopsies. External validation of this MRI-PM has not been done, and no RC has been designed.

In 2021, Sakaguchi et al. [35] developed an MRI-PM in 773 biopsy-naïve men with suspected PCa recruited in a single centre in Tokyo, Japan. Pre-biopsy bpMRI was reported with PI-RADS v.2, and 2- to 4-core guided-cognitive biopsies and/or from 8- to 14-core systematic biopsies were carried out through a transrectal approach. The MRI-PM shared age, serum PSA, and PV as clinical predictors of csPCa, defined as GS > 4+3 or maximum cancer core length > 6 mm, and 44.3% of csPCa was detected. The AUROC of MRI alone (0.82) increased to 0.86 when MRI and clinical predictors were combined. DCAs showed a net benefit for MRI-PM over MRI from the csPCa risk threshold of 35%. At 95% sensitivity, the MRI-PM presented 73% specificity, and 43% of prostate biopsies were avoided. This model was not externally validated, and no RC was designed.

Finally, in 2022 two new MRI-PMs have been reported. Kinnaird et al. [17] developed an MRI-PM in 2354 men with suspected PCa in whom mpMRI was reported by PI-RADS v2 and 2- to 3-core cognitive-guided biopsies and/or 12-core systematic biopsy by transrectal approach were performed between 2009 and 2019 in two centres in Los Angeles and New York, US. The rate of csPCa detected was 40%. The authors divided the po- pulation into five groups, and one of them was randomly selected for internal validation. The developed MRI-PM shared the clinical predictors, including age, DRE, biopsy status, ethnicity, serum PSA, PSAD, and PV. The AUROC of MRI-PM (0.84) outperformed that of MRI alone (0.76). DCAs evaluating the net benefit were not provided. At 95% sensiti-vity, the specificity of the MRI-PM was 56%, and 40% of prostate biopsies were avoided. External validation of this model has not been performed, and a designed RC is not currently available. Last, Morote et al. [16] designed the Barcelona RC from an MRI-PM developed in 1486 men with PSA > 3 ng/mL or abnormal DRE. The developed model was externally validated in a cohort of 946 men belonging to two centres in the same metropolitan area. The rate of csPCa, defined when GG > 2, in the development cohort was 36.9% and 40.8% in the validation cohort. The developed model shared the independent predictors of age, PCa family history, biopsy status, DRE, serum PSA, PV, and PI-RADS v2. Prostate biopsies were performed by transrectal approach with two to four-core cognitive-guided biopsies to suspicious lesions and/or twelve-core systematic biopsy. The AUROC of MRI-PM increased that of MRI from 0.84 to 0.90 (*p* = 0.01) in development cohort and from 0.74 to 0.86 in validation cohort (*p* < 0.001). At the 15% csPCa threshold, 40.1% of overall biopsies would be avoided, and 95.7% of csPCa would be detected in the development cohort, while the specificity was 39.9% and 89.5% would be avoided in the validation cohort. This MRI-PM was analysed according to the PI-RADS categories for the first time. The authors concluded after this analysis that the clinical utility and net benefit of the Barcelona MRI-PM in entire developed and validation cohorts did not represent the true usefulness of the model regarding PI-RADS categories. The model’s performance was low for men with PI-RADS >3, especially those with PI-RADS 5. The performance and clinical utility were greater in men with PI-RADS <3; in those men with PI-RADS 3, 61.9% and 62.3% of prostate biopsies would be avoidable in development and validation cohorts while 28.2% and 14% of the csPCa would be missed. In men with negative mpMRI (PI-RADS < 3), 4.3% and 6.5% of biopsies showed that 33.3% and 18.2% of existing csPCa would be detected in development and validation cohorts, respectively.

Many previous studies have compared baseline PMs based on clinical predictors, MRI alone, and the MRI-PM [17,21,22,26,28,29,32,34]. Other studies have compared the MRI-PM with MRI alone [15,17,25,26,32,33,34,35]. AUROCs for MRI setting alone, PM based only in clinical predictors and MRI-PMs are collected in Table 4. Net benefit was often represented by DCAs [15,16,17,26,27,28,29,30,31,32,33,35]. Some studies analysed the performance of MRI-PM from selected thresholds [24,28,29,32] while few studies reported CUCs, which represent the continuous probabilities of avoided prostate biopsies and missed csPCa regarding the continuous risk threshold of csPCa [15,16]. Only a single developed MRI-PM was analysed according to each PI-RADS category [16].

## 4. Discussion

Predictive models are the only tools providing the individualised risk of csPCa of men with suspected PCa [40]. These models have evolved and currently share MRI findings, mainly reported through PI-RADS v2, and independent clinical predictors helping clinicians improve the selection of candidates for prostate biopsy. This strategy can reduce unnecessary prostate biopsies in uncertain scenarios beyond avoiding those in negative MRIs (PI-RADS < 3), while missing acceptable rates of csPCa according to the incidence in PI-RADS categories and its aggressiveness [13]. Almost all current MRI-PMs analysed in this systematic review have shown increased discrimination ability for csPCa than baseline models based on clinical predictors without MRI [28,31] and MRI alone [34,35].

Considerations of MRI characteristics are important since it is the weightiest predictor of csPCa [41]. Both 1.5 and 3 Tesla MRIs have been used for developing the reported MRI-PMs; however, the most recently developed models used a 3 Tesla MRI. The acquisition protocol of mpMRI included T2-weighted imaging (T2W), diffusion-weighted imaging (DWI) and dynamic contrast-enhanced (DCE) imaging [42]. In contrast, bpMRI, which avoids dynamic contrast-enhanced (DCE) imaging, is a cheaper and faster procedure than mpMRI while maintaining high diagnostic accuracy [43,44,45]; these were incorporated in MRI-PMs developed in high volume centres with experienced radiologists [24,32,34,35], even though comparison data between both MRI techniques in PMs is still missing. Update PI-RADS version 2 were used exclusively in eleven of the eighteen developed MRI-PMs [15,16,17,25,27,30,31,32,34,35,37]. A recent meta-analysis found higher pooled sensitivity of PI-RADSv2 over PI-RADSv1, but with similar pooled specificity [46]. In an external validation of the Radtke’s MRI-PM based on PI-RADSv1 improved performance was observed when PI-RADSv1 was replaced by PI-RADSv2 [47]. However, consistent data analysing the influence of the PI-RADS version on MRI-PMs performance is lacking.

Clinical predictors incorporated in developed MRI-PMs are those with significant and independent weight to predict csPCa in logistic regression analysis [48]. Although PSA has known limitations it is still the mainstay for establishing PCa suspicion [49] and all MRI-PMs include serum PSA alone or in combination with prostate volume as PSAD [15,17,23,24,25,32,34]. In two of the analysed MRI-PMs [21,32], men with serum PSA levels above 50 ng/mL were excluded from the development cohort. Due to the low specificity of serum PSA between 4 and 10 ng/mL [50], Niu et al. developed an MRI-PM in patients with serum PSA in the “grey zone” in whom it could not predict csPCa. Contrarily, MRI-PM predicted csPCa efficiently in this uncertain scenario. [25]. PSAD has gained prominence since the spread of MRI, which provides the assessment of PV without additional cost and avoids the annoying and time-consuming training of urologists for transrectal ultrasonography (TRUS) [41]. Moreover, PV estimated volume by MRI either measured by ellipsoid or bullet formula has been shown to be the most accurate method compared to TRUS and DRE [51]. PSAD is the most powerful predictor of csPCa after the PI-RADS score [52]. PSAD has shown its dynamic behaviour across the PI-RADS categories, which improves its efficacy after selecting the appropriate threshold in each PI-RADS [12]. Age is related to a higher incidence and aggressiveness of PCa [53]. The Rotterdam MRI-RC and the MRI-PM developed by Boesen et al. restricted the age from 50 to 74 years, limiting clinical practice [32]. Although DRE is questioned due to the modest incremental gain of csPCa detection in men with low serum PSA [54], abnormal DRE is associated with increased detection of higher-grade tumours [55]. DRE has been commonly incorporated in developed MRI-PMs [15,16,17,21,22,26,28,29,30,31,32]. However, Niu et al. [25] and Truong et al. [27] did not include DRE in their MRI-PMs because it did not result in an independent predictor of csPCa in their logistic regression analysis. In the remaining MRI-PMs, DRE was not reported or not included in logistic regression analyses [23,24,34,35,37]. Ethnicity was only included as a predictor in three MRI-PMs [17,22,31]. In some developed MRI-PMs, ethnicity was not considered [15,23,24,26,29]; in others, it was not included as a predictor due to the low heterogenicity of race distribution, e.g. cohorts based in Asian [21,25,30,34] or Caucasian [28,32] populations. Kinnaird et al. [17] noted that Asian ethnicity predicted a reduced risk of csPCa in the American population, and this observation is consistent with the available literature [56]. Black men have been shown to have a poorer PCa prognosis and a higher risk of recurrence than white men [57]. However, a recent multiple-cohort study have seen no differences of PCa specific mortality in Black population after the adjustment of nonbiological differences [58]. Previous negative prostate biopsy is a common occurrence in csPCa early detection. Four MRI-PMs have been developed only with biopsy-naïve men [21,25,32,35]. MRI-PMs based only on one type of biopsy status restricts their usefulness in everyday practice [21,25,27,30,32,34,35].

The biopsy scheme influence csPCa detection rate and its heterogenicity was observed in this systematic review. Between 8 to 14-core systematic prostate biopsy associated to 1 to 4-core guided biopsies was reported in nine developed MRI-PMs [15,16,17,23,25,30,31,32,35]. Some MRI-PMs did not provide information about the number of cores in guided biopsies [21,22,29]. 24- to 40-cores template biopsies and 2- to 4-core guided biopsies were reported in three developed MRI-PMs [24,26,28]; last, in one developed MRI-PM, the biopsy scheme was not reported [30], although in one it was previously described [34]. All biopsy approaches were transrectal except those using template biopsies, which used the transperineal approach [24,28,34]. Currently, studies comparing the influence of biopsy schemes on the efficacy of developed MRI-PM are lacking. Since 2021, the European Association of Urology PCa guidelines has highly recommended performing prostate biopsies using a transperineal approach due to the lower risk of infectious complications [59]. Therefore, MRI-PMs developed with biopsy schemes using the transrectal approach should be validated with the same schemes but carried out through a transperineal approach. A recent meta-analysis exhibited 86% csPCa detection rate when a transperineal approach was used against 73% when a transrectal approach [60]. The possibility of targeting biopsies to suspicious lesions detected in MRI increases the sensitivity of prostate biopsy to detect csPCa [10]. However, the Cochrane metanalysis noted the complementarity between guided and systematic biopsies [7]. All developed MRI-PMs incorporated cohorts of men in whom a cognitive approach (visual fusion between MRI and TRUS images) or software approach (MRI-TRUS fusion image with software) were performed. In one study, both types of guided biopsies and in-bore guided biopsies were performed [29]. Eight studies used exclusively cognitive guided biopsies [15,16,17,24,25,32,34,35], four only used software-guided biopsies [23,26,27,31] and two studies used both types of guided biopsies [22,28]. Based on the current knowledge, there is no clear benefit of software-guided biopsies over cognitive-guided biopsies, especially when experienced practitioners [61]. A recent meta-analysis regarding guided biopsies has shown similar csPCa detection rates for cognitive-guided biopsies, software-guided biopsies, and in bore-guided biopsies [62].

The definition of csPCa as the main outcome of MRI-PMs also has changed in deve- loped MRI-PMs. In four MRI-PMs an ISUP-GG > 2 was the used definition [15,16,17,32]. In eleven MRI-PMs Gleason grade 3+4 or higher was used [21,22,23,25,26,27,29,30,31,33,34]. Three MRI-PMs used definitions based on Gleason grade 3+4 and higher than 6–7 mm cancer core extension [24,28,35]. To our knowledge, no MRI-PM has been developed with the recommendation of the 2019 Consensus Conference on Grading Prostatic Carcinoma, which considers with poor prognosis the GG 2 tumours with cribriform differentiation or intraductal carcinoma [63]. Today, the Rotterdam RCs consider csPCa when GG > 2 or GG = 2 with cribriform differentiation or intraductal carcinoma, which was applied to RC-3 and RC-4 but not to the Rotterdam MRI-RC [39].

All included MRI-PMs use as a reference standard the histopathological findings of csPCa in the prostate biopsies. Liu et al. [64] developed a model with PI-RADSv2, PSA and PSAD for the prediction of PCa and csPCa for low and high-risk groups. In this study, histopathological findings of PCa were obtained from prostate biopsy but also from the whole prostate gland, including transurethral resection of the prostate, transurethral enucleation of the prostate and radical prostatectomy. Limitations of a PM based on a cohort of patients whose pathological specimens were obtained by a surgical approach are the exclusion of patients who underwent radiotherapy, active surveillance, watchful waiting, metastatic patients or refusal of standard of care. However, strengths could be the presence of the histology of the whole prostate gland, as prostate a biopsy may downgrade prostate cancer [65].

External validations are the key point for the extensive use of MRI-PMs in the deve- lopment population. However, external validations that guarantee accurate predictions have been made in less than half of developed MRI-PMs [16,27,28,29,31,33,34]. The significant difficulty of external validations of predictive models is the difference between development incidence and validation cohorts. Predictive models will overestimate csPCa likelihood when the incidence decreases and underestimate csPCa risk when the incidence increases [66]. An adjusted selection of threshold probability of csPCa might be required to sidestep this limitation [67]. Mehralivand et al. [31] validated their model in a cohort of 251 men from two independent centres. Despite a difference of about 10% in csPCa incidence between the development and validation cohort, the AUROCs were similar; however, after adjusting the probability threshold of csPCa to 20%, 38% of prostate biopsies were saved while missing 11% of csPCa. Pullen et al. [68] compared the external validation of three MRI-PMs in a cohort of 307 men. The AUROCs of the three models were similar; however, after adjusting their risk threshold, better calibration and net benefit of the Rotterdam MRI-RC were observed in biopsy-naïve men over the other two models with a worse balance between saved prostate biopsies and missed csPCa. Saba et al. [69] validated and compared three MRI-PMs [26,28,31], the Rotterdam MRI-RC [29] and the Prostate Biopsy Collaborative Group (PBCG)-RC [70] in 468 suspected PCa men, 31% with previous negative prostate biopsy, in whom transperineal 39 to 42-core template biopsy with additional 2 to 3-core software or cognitive-guided biopsies were performed between 2014 and 2018 in Zürich, Switzerland. The finding of any Gleason pattern 4 in prostate biopsy was considered csPCa. The authors concluded that MRI-PMs outperformed those models without MRI. However, the clinical net benefit was only observed when the risk threshold was not greater than 15%.

Risk calculators (RCs) are essential for the routine use of MRI-PMs because they avoid the cumbersome and time-consuming use of nomograms [38]. However, only three MRI-RCs were designed [15,16,17], and two of them are available [15,16]. The prestigious Rotterdam MRI-RC was validated by De Nunzio et al. [38] in a multi-centre study on 580 suspected PCa men from 12 Italian centres. Recalibration was necessary to obtain a net benefit between 20% and 80% csPCa probability threshold. More recently, Remmers et al. validated the Rotterdam MRI-RC in the PRECISION Trial population that only included biopsy-naïve men. Recalibration and adjustment of the csPCa probability threshold were also necessary to reach appropriate performance, which avoided 28% of prostate biopsies while missing 10% of csPCa when the 10% risk threshold for csPCa was used [39]. The Barcelona RC was designed from a developed MRI-PM using the same variables as the Rotterdam MRI-RC and PCa family history [16]. This new MRI-PM was produced in almost 1500 suspected PCa men in whom a 3 Tesla pre-biopsy mpMRI reported with PI-RADSv2 and the recommended by the EAU PCa guidelines scheme of guided and systematic prostate biopsy. The definition of csPCa was the same as the Rotterdam MRI-RC, as was the biopsy scheme and approach [29]. The Barcelona MRI-PM was externally validated in almost 1000 men from the same metropolitan area. The Barcelona RC was designed with the new option of selecting the threshold of csPCa risk, which offers the possibility of facilitating further external validations [16]. The Barcelona MRI-PM and RC were analysed according to the PI-RADS categories; their behaviour showed how the performance in the overall population did not reflect the performance in each PI-RADS category. Therefore, this analysis has shown the scenarios where the MRI-PM is helpful and what the optimal threshold to be used in each PI-RADS category based on the acceptable rate of missing csPCa is, according to its incidence and aggressiveness of detected tumours [16].

To summarise the synthesis of generated evidence from this systematic review, we note that MRI-PMs are the best way to individualise the risk of csPCa among men with suspected PCa. However, available MRI-RCs are essential to predict the individual csPCa risk in routine practice. Although the overall risk of bias of selected articles for this review was low, developed MRI-PMs are heterogeneous regarding the characteristics of development cohorts, the characteristics of MRIs scanners and reporting methods, the clinical predictors included in the models, the types of prostate biopsy schemes and approaches, and the definitions of csPCa. Few external validations were carried out, and recalibrations of the model and adjustment of csPCa risk threshold were necessary when accurate predictions were made. The analysis of MRI-PMs regarding PI-RADS categories seems reasonable because clinical usefulness in overall populations does not represent the true utility in each PI-RADS. However, this analysis has only been carried out in a single MRI-PM, and, incredibly, only two RCs are currently available.

There is an emerging need for tools to improve the radiologist interpretation of MRI. Radiomics, known as the conversion of digital images into mineable high-dimensional data, which has gradually become a research focus in recent years [71]. uture developments envisage a comparison between radiomics performance and the radiologist eye [72]. There are existing algorithms that can detect suspicious lesions missed by the radiologist [73] and to distinguish accurately clinically significant lesions from indolent lesions and normal tissue in MRI [73,74]. These innovative tools identify histological grades of aggressiveness in suspicious lesions by a pixel-level label for the whole-mount histopathology, which can overstep the limitation of PMs based on biopsy histopathology. Moreover, other radiomic models have been proved to improve some aspects of the performance of PI-RADS v2 in predicting of csPCa [75,76,77]. Deep learning techniques might also make a difference in everyday clinical practice being used for prostate gland segmentation, increasing its speed and reducing error for suspicious lesion identification [78]. Mentioned strategies might help to improve csPCa detection by assisting guided targeted biopsies, reducing unnecessary biopsies while not missing csPCa. Radiomics will improve the future prediction of PI-RADS, and genomics will enhance the definition of csPCa [79]. The integration of big data from radiogenomics and clinical predictors using artificial intelligence algorithms will improve the prediction true csPCa. Because differences in csPCa incidence will persist among populations with suspected PCa, external validations of the generated MRI-PMs will be unresolved unless a federated network is implanted [79]. Recent appearance of this artificial intelligence algorithms create a need for validation tools such as ProstateX and the new Prostate Image: Cancer AI (PI-CAI) training (https://pi-cai.grand-challenge.org/) (accessed on 27 September 2022). This available dataset provides clinical information that could also be used for AI-based PM development. The continuous update of an MRI-RC by machine learning algorithms and feedback through the new diagnosed cases will provide the models with the evolution of attended populations, the current diagnostic approaches, and csPCa definition refinements to maintain the effectiveness of csPCa predictions [80].

## 5. Conclusions

MRI-PMs are the current way to individualise the risk of csPCa and improve the selection of candidates for prostate biopsy. Available RCs are essential to facilitate the routine use of MRI-PMs. External validations are needed to assure accurate predictions of csPCa in any population where a developed MRI-PM is to be accomplished. Among the recently developed MRI-PMs, few meet the appropriate requirements used in routine clinical practice. In addition, only a single MRI-PM has been analysed according to the PI-RADS categories, which seems essential to define the scenarios of clinical usefulness and the appropriate csPCa risk thresholds. Future MRI-PMs should incorporate improved methods of prediction through radiomics, better definition of csPCa through genomics, and the integration of bigdata and the generation of artificial intelligence algorithm. The key point of external validations should be resolved through the integration and feedback of generated algorithms in federated networks.

## Figures and Tables

**Figure 1 cancers-14-04747-f001:**
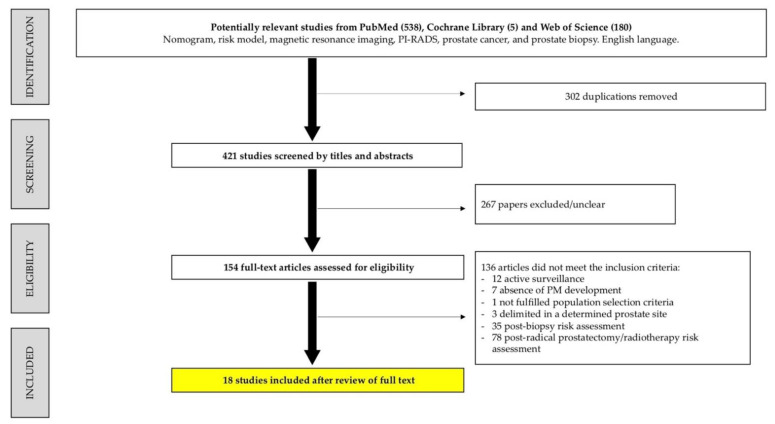
Flow chart of systematic review according to the PRISMA criteria.

**Figure 2 cancers-14-04747-f002:**
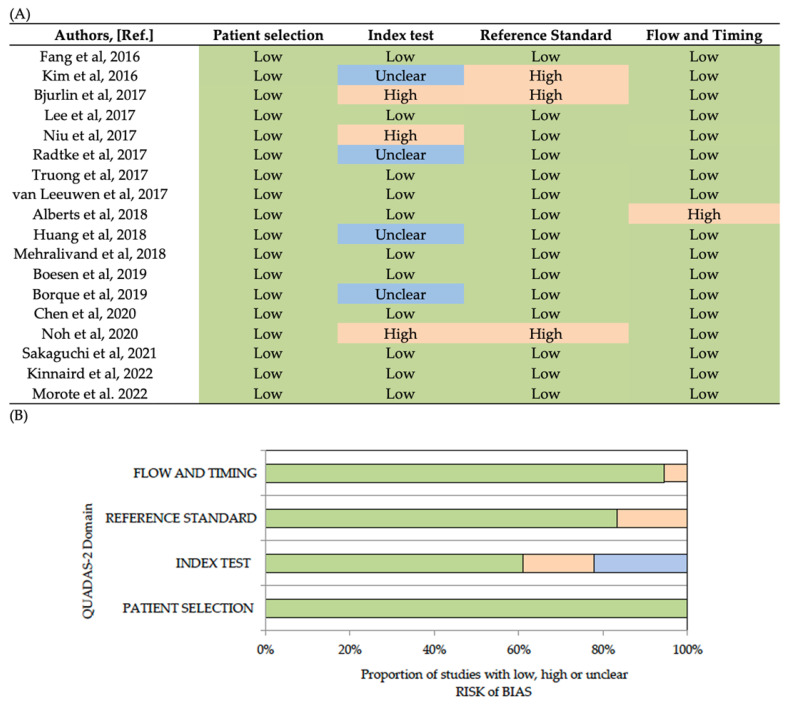
Analysis of bias risks (low, unclear, high) of analysed studies according to each bias. domain (**A**), and proportion of studies according to each domain (**B**) [15,16,17,21,22,23,24,25,26,27,28,29,30,31,32,33,34,35,36].

**Table 1 cancers-14-04747-t001:** Biopsy status and size of development cohorts, characteristics of pre-biopsy MRI, prostate biopsy approaches and types of systematic and guided biopsies, and csPCa definitions.

Authors, Year [Ref.]	Biopsy Status	MRI/T	PI-RADS Version	Biopsy Approach	Systematic Biopsy	Guided Biopsy	Type of GB	csPCa Definition
Fang et al., 2016 [21]	984/0	mp/1.5–3	1	TR	12	NA/≥3	NA	GS ≥ 3+4
Kim et al., 2016 [22]	185/154	mp/3	1–2	TR	12	NA/≥4	Cog/Soft	GS ≥ 3+4
Bjurlin et al., 2017 [23]	288/171	mp/3	1	TR	12	1–4/≥3	Soft	GS ≥ 3+4
Lee et al., 2017 [24]	484/131	bp/1.5	1	TP	24–40 *	2–4/≥3	Cog	GS≥7 or MCCL≥ 6 mm
Niu et al., 2017 [25]	151/0	mp/3	2	TR	12	1/≥3	Cog	GS ≥ 3+4
Radtke et al., 2017 [26]	670/489	mp/3	1	TP	24 *	2–4/≥2	Soft	GS ≥ 3+4
Truong et al., 2017 [27]	0/285	mp/3	2	TR	12–24 *	2/≥3	Soft	GS ≥ 3+4
van Leeuwen et al., 2017 [28]	344/49	mp/1.5–3	1	TP	30 *	2/≥3	Soft/Cog	GS≥ 7/> 5% G4 or MLCL≥ 20%/7 mm
Alberts et al., 2018 [29]	504/457	mp-bp/3	1–2	TR	12	NA/≥3	In bore/Cog/Soft	GS ≥ 3+4
Huang et al., 2018 [30]	0/231	mp/1.5–3	2	TR	12	2/≥4	NA	GS ≥ 3+4
Mehralivand et al., 2018 [31]	179/221	mp/NA	2	TR	12	2/≥3	Soft	GS ≥ 3+4
Boesen et al., 2019 [32]	876/0	bp/3	2	TR	10	2/≥3	Cog	GG ≥ 2
Borque et al., 2019 [15]	163/183	mp/3	2	TR	12	2/≥3	Cog	GG ≥ 2
Chen et al., 2020 [33]	316	mp/NA	2	NA	NA	NA	NA	GS ≥ 3+4
Noh et al., 2020 [34]	215/85	bp/3	2	TP	24–20 *	2–10/≥3	Cog	GS ≥ 3+4
Sakaguchi et al., 2021 [35]	773/0	bp/1.5–3	2	TR	8–14	2–4/≥3	Cog	GG3 or MCCL≥ 6 mm
Kinnaird et al., 2022 [17]	1449/905	mp/3	2	TR	12	2–3/≥3	Cog	GG ≥ 2
Morote et al. 2022 [16]	1098/388	mp/3	2	TR	12	2–4/≥3	Cog	GG ≥ 2

Ref. = reference; Biopsy status = number of biopsy naïve men/number of repeat biopsy; MRI = magnetic resonance imaging (mp = multiparametric, bp = biparametric/T = Tesla); Biopsy approach = TR (Transrectal), TP (Transperineal); Systematic biopsy = number of cores (* template); Guided biopsy = guided biopsy (Soft: software, Cog: cognitive); GB = guided biopsy; csPCa = clinically prostate cancer; GS = Gleason score; GG = grade group; MCCL = maximal core cancer length; NA = not available.

**Table 2 cancers-14-04747-t002:** Clinical predictors included in developed MRI-PMs for clinically significant prostate cancer.

Authors, [Ref.]	Age	PCa FH	DRE	Biopsy Status	Ethnicity	PSA	PSAD	PV
Fang et al., 2016 [21]	Y	N	Y	N	N	Y	N	Y
Kim et al., 2016 [22]	Y	Y	Y	Y	Y	Y	N	N
Bjurlin et al., 2017 [23]	Y	N	N	N	N	N	Y	N
Lee et al., 2017 [24]	Y	N	N	Y	N	N	Y	N
Niu et al., 2017 [25]	Y	N	N	N	N	N	Y	N
Radtke et al., 2017 [26]	Y	N	Y	N	N	Y	N	Y
Truong et al., 2017 [27]	Y	N	N	N	N	Y	N	Y
van Leeuwen et al., 2017 [28]	Y	N	Y	N	N	Y	N	Y
Alberts et al., 2018 [29]	Y	N	Y	N	N	Y	N	Y
Huang et al., 2018 [30]	Y	N	Y	N	N	Y	N	Y
Mehralivand et al., 2018 [31]	N	N	Y	Y	Y	Y	N	N
Boesen et al., 2019 [32]	Y	N	Y	N	N	N	Y	N
Borque et al., 2019 [15]	Y	N	Y	Y	N	N	Y	N
Chen et al., 2020 [33]	N	N	N	N	N	Y	N	Y
Noh et al., 2020 [34]	Y	N	N	N	N	N	Y	N
Sakaguchi et al., 2021 [35]	Y	N	N	N	N	Y	N	Y
Kinnaird et al., 2022 [17]	Y	N	Y	Y	Y	Y	Y	Y
Morote et al. 2022 [16]	Y	Y	Y	Y	N	Y	N	Y

Y = yes; N = no; PCaFH = prostate cancer family history; DRE = digital rectal examination; PSA = prostate-specific antigen; PSAD = PSA density; PV = prostate volume.

**Table 3 cancers-14-04747-t003:** Clinical usefulness of developed MRI-PMs.

Authors, [Ref.]	n	Repeat Biopsy	csPCa	Sen.	Spe.	Avoided Biopsies	Cut-Off	AUROC	DCA	CUC
Fang et al., 2016 [21]	894	0	24.4	95	38	19.8	30	0.87	5	NA
Kim et al., 2016 [22]	339	35.4	34.0	95	20	15.1	NA	0.78	NA	NA
Bjurlin et al., 2017 [23]	288	0	33.6	95	56	42.2	NA	0.91	NA	NA
Bjurlin et al., 2017 [23]	171	100	18.1	95	40	33.9	NA	0.86	NA	NA
Lee et al., 2017 [24]	615	21.3	38.5	97.5	54.8	34.6	30	0.92	NA	NA
Niu et al., 2017 [25]	151	0	21.0	87.3	78.4	64.9	36	0.85	NA	NA
Radtke et al., 2017 [26]	660	0	NA	95	35	NA	NA	0.83	16	NA
Radtke et al., 2017 [26]	335	100	NA	95	25.5	NA	NA	0.81	12	NA
Truong et al., 2017 [27]	285	100	38.9	94.7	57.5	36.5	40	0.83	1	NA
van Leeuwen et al., 2017 [28]	393	12.5	37.9	93.9	NA	34.4	12.5	0.88	4	NA
Alberts et al., 2018 [29]	504	0	42.0	92	NA	24.0	15	0.84	10	NA
Alberts et al., 2018 [29]	504	100	29.0	95	NA	41.0	15	0.85	5	NA
Huang et al., 2018 [30]	231	100	25.5	95	63	48.0	21	0.92	10	NA
Mehralivand et al., 2018 [31]	400	55.2	48.3	96	54	30.0	15	0.84	10	NA
Boesen et al., 2019 [32]	876	0	40.0	96	60	38.0	15	0.89	5	NA
Borque et al., 2019 [15]	346	53.0	32.6	95	51	30.0	10	0.88	0.88	Y
Chen et al., 2020 [33]	257	NA	59.2	95	40	19.0	NA	0.84	NA	NA
Noh et al., 2020 [34]	300	28.3	34.0	95	52	30.1	10	0.86	10	NA
Sakaguchi et al., 2021 [35]	773	0	44.3	95	73	43.0	15	0.86	5	NA
Kinnaird et al., 2022 [17]	1885	62.0	40.0	95	32	21.2	NA	0.84	NA	NA
Morote et al. 2022 [16]	1486	26.1	36.9	95	56	40.0	15	0.90	12	Y

n = number of men; RB = percentage of repeat biopsies; csPCa = percentage of clinically significant prostate cancer; Sen = sensitivity; Spe = specificity; Repeat biopsy = percentage; Sen. = percent sensitivity; Esp. = percent specificity; Avoided biopsies = percentage; AUROC = area under Receiver operating characteristic curve; DCA = decision curve analysis; CUC = clinical utility curve; NA = not available.

**Table 4 cancers-14-04747-t004:** AUROCs for MRI setting alone, PM based only in clinical predictors and MRI-based PMs.

	AUROC for csPCa		
Authors, Year [Ref.]	MRI Setting Alone	Clinical Predictors Predictive Model	MRI-Based Predictive Model
Fang et al., 2016 [21]	NA	BN: 0.85 PNPB: NA Both status: NA	BN: 0.872 PNPB: NA Both status: NA
Kim et al., 2016 [22]	NA	BN: 0.60 PNPB: 0.63 Both status: 0.60	BN: 0.72 PNPB: 0.61 Both status: 0.69
Bjurlin et al., 2017 [23]	NA	NA	BN: 0.84 PNPB: 0.87 Both status: NA
Lee et al., 2017 [24]	NA	NA	BN: NA PNPB: NA Both status: 0.92
Niu et al., 2017 [25]	BN: 0.76 PNPB: NA Both status: NA	NA	BN: 0.85 PNPB: NA Both status: NA
Radtke et al., 2017 [26]	BN: 0.76 PNPB: 0.78 Both status: NA	BN: 0.81 PNPB: 0.66 Both status: NA	BN: 0.83 PNPB: 0.81 Both status: NA
Truong et al., 2017 [27]	NA	NA	NA
van Leeuwen et al., 2017 [28]	NA	BN: NA PNPB: NA Both status: 0.797	BN: NA PNPB: NA Both status: 0.897
Alberts et al., 2018 [29]	NA	BN: 0.76 PNPB: 0.74 Both status: NA	BN: 0.84 PNPB: 0.85 Both status: NA
Huang et al., 2018 [30]	NA	NA	BN: NA PNPB: 0.927 Both status: NA
Mehralivand et al., 2018 [31]	NA	BN: NA PNPB: NA Both status: 0.72	BN: NA PNPB: NA Both status: 0.84
Boesen et al., 2019 [32]	BN: 0.83 PNPB: NA Both status: NA	BN: 0.85 PNPB: NA Both status: NA	BN: 0.89 PNPB: NA Both status: NA
Borque et al., 2019 [15]	NA	NA	BN: NA PNPB: NA Both status: 0.856
Chen et al., 2020 [33]	0.869	NA	0.84
Noh et al., 2020 [34]	BN: 0.801 PNPB: NA Both status: NA	BN: 0.795 PNPB: NA Both status: NA	BN: 0.861 PNPB: NA Both status: NA
Sakaguchi et al., 2021 [35]	BN: 0.822 PNPB: NA Both status: NA	NA	BN: 0.862 PNPB: NA Both status: NA
Kinnaird et al., 2022 [17]	BN: NA PNPB: NA Both status: 0.760	BN: NA PNPB: NA Both status: 0.707	BN: NA PNPB: NA Both status: 0.843
Morote et al. 2022 [17]	BN: NA PNPB: NA Both status: 0.842	NA	BN: NA PNPB: NA Both status: 0.987

AUROC = area under Receiver operating characteristic curve; csPCa = clinically significant prostate cancer; MRI = magnetic resonance imaging, BN = biopsy-naïve, PNPB = previous negative prostate biopsy, NA = not available.

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
