# Peer review of "Magnetic Resonance Imaging-Based Predictive Models for Clinically Significant Prostate Cancer: A Systematic Review"

_cancers, 2022, doi:10.3390/cancers14194747_

Round 1

Reviewer 1 Report

COMMENT TO AUTHORS

The current study aims at analysing the current developed MRI-PMs and define their clinical usefulness.

The study demonstrates that MRI-PMs improve the selection of candidates for prostate biopsy beyond the PI-RADS category. However, few developed MRI-PMs meet the appropriate requirements to be used in routine clinical practice.

The authors should be congratulated for the work and for addressing an important topic. Only few points warrant mentions:

Major comments:

1. In “Introduction” section, the definition provided by the authors of clinically significant PCa is not clear. I suggest to better define this definition.

Minor comments:

1.    In the “Discussion” section, I suggest completing the information regarding the MRI predictive models including more information on Radiomics as in PMID: 35814914 “Radiomics in prostate cancer: an up-to-date review”.

2.    In the “Discussion” section, the authors provide information on the clinical importance of prostate volume. I suggest to cite also PMID: 34247169 “The Comparison of Imaging and Clinical Methods to Estimate Prostate Volume: A Single-Centre Retrospective Study”, in this section.

Author Response

Dear Reviewer,

We would like to thank you and the other reviewers for your comments and suggestions. These have greatly helped us to improve the quality of our manuscript.

Find in attachement the changes according to your contributions below.

Reviewer 2 Report

This paper systematically reviews papers regarding MRI-based predictive models for clinically significant prostate cancer.

1) It appears that Simple Summary includes only introductory content and Abstract includes methods, results, and conclusion. I think that Simple Summary should be a simple summary of this paper. In addition, introduction and aim should be included in Abstract.

2) In all 18 papers, AUROCs of predictive models including clinical parameter-based models, MRI parameter-based models, and clinical and MRI parameter-based models, should be summarized in the Table. 

3) Does reviewed MRI-PMs include not only PIRADS category but also other MRI parameters such as tumor contact length? If so, they should be written in the Table.

4) It is clinically important to increase the predictive ability for detecting csPCa in patients with PIRADS category 3. What clinical parameters can be useful in patients with PIRADS category 3? This should be discussed.

Author Response

Dear Reviewer,

We would like to thank you and the other reviewers for your comments and suggestions. These have greatly helped us to improve the quality of our manuscript.

 Find attached changes according to your contributions.

Reviewer 3 Report

Thank you for the opportunity to review the manuscript entitled “Magnetic Resonance Imaging-Based Predictive Models for Clinically Significant Prostate Cancer: A Systematic Review.”. The clinical topic is essential, as the potential clinical use of advanced predictive methods could help optimize the diagnosis of prostate cancer, the most common cancer in men. However, I have several comments to improve the quality of the manuscript.

  1. Certain parts of the simple summary and abstract can be difficult to read. E.g., what does sharing the PIRADS score mean? Or sharing predictors to PIRADS scores? PIRADS scores are scores radiologists use to classify lesions before a biopsy. However, they might over- or underestimate the extent of cancer.
  2. A concern is that the authors do not split up the information between predictors for biopsy cohorts (both men with and without cancer) and radical prostatectomy cohorts (everybody had cancer). It would be interesting if papers were subcategorized, possibly as a sensitivity analysis.
  3. Could the authors investigate cohorts that compared race/ethnicity? Especially information on studies that included persons of Black race/ethnicity would be interesting, as they have shown to be of higher risk.
  4. Many recent papers focusing solely on deep learning for prostate imaging by Rusu et al. have been left out that warrant being added to the review, possibly in the introduction or discussion sections. These include papers on detection of both aggressive and indolent prostate cancer (https://pubmed.ncbi.nlm.nih.gov/33760269/), detection using correlated learning (https://pubmed.ncbi.nlm.nih.gov/34784540/), and prospective AI utilization for gland segmentation which is needed for optimal registration and post-processing (https://pubmed.ncbi.nlm.nih.gov/33878887/). This study was also utilized prospectively. 
  5. Moreover, it could be interesting to see how many studies utilized biopsy-verified lesions as inputs. E.g., in most clinics, radiologists annotate the lesions initially. Those lesions can be saved and used as labels. However, it would be more precise if those lesions were verified by biopsy or whole-mount histopathology.
  6. It would benefit the paper if the authors touched on publicly available datasets, e.g., ProstateX or the newer PI-CAI, and whether this kind of data should be utilized. E.g., PI-CAI also has clinical variables such as age and PSA values.
  7. It would also be relevant if the authors looked into whether papers analyzed information with lesion-level information or whether it was patient-level.
  8. Could the authors mention pros as cons (limitations) to the final studies included? Possibly as an extra column in an overview table?

Author Response

Dear Reviewer,

We would like to thank you and the other reviewers for your comments and suggestions. These have greatly helped us to improve the quality of our manuscript.

Find in attachement changes according to your contributions .  

Round 2

Reviewer 1 Report

Authors answered all comments and suggestions.

Reviewer 3 Report

The authors have done a nice job responding to my comments. The revised paper is much easier for me to follow. I do not have additional comments.